# Beta-Genus Human Papillomavirus 8 E6 Destabilizes the Host Genome by Promoting p300 Degradation

**DOI:** 10.3390/v13081662

**Published:** 2021-08-21

**Authors:** Dalton Dacus, Nicholas A. Wallace

**Affiliations:** Division of Biology, Kansas State University, Manhattan, KS 66506, USA; dacus@ksu.edu

**Keywords:** HPV, skin cancer, p300, DNA damage, proliferation, differentiation, UV

## Abstract

The beta genus of human papillomaviruses infects cutaneous keratinocytes. Their replication depends on actively proliferating cells and, thus, they conflict with the cellular response to the DNA damage frequently encountered by these cells. This review focus on one of these viruses (HPV8) that counters the cellular response to damaged DNA and mitotic errors by expressing a protein (HPV8 E6) that destabilizes a histone acetyltransferase, p300. The loss of p300 results in broad dysregulation of cell signaling that decreases genome stability. In addition to discussing phenotypes caused by p300 destabilization, the review contains a discussion of the extent to which E6 from other β-HPVs destabilizes p300, and provides a discussion on dissecting HPV8 E6 biology using mutants.

## 1. Introduction

The stability of our genome is threatened by internal and external hazards [1]. Endogenous sources of genomic destabilization include errors during replication and mitosis, as well as reactive oxygen species resulting from metabolism [2,3]. Exogenous sources are equally prevalent, including ultraviolet radiation (UV), ionizing radiation (IR), and naturally occurring as well as human-made mutagens [4,5]. To minimize the mutations associated with these genotoxic events and agents, cells have evolved an elaborate collection of specific DNA repair and cell cycle arrest signaling pathways. Collectively they are known as the DNA damage response (DDR) and play an integral part in maintaining genome stability [6,7]. When necessary, DDR signaling can also include the induction of apoptosis [8]. The DDR has exquisite specificity as individual pathways specializing in a single type of damage/challenge occurring in one portion of the cell cycle [9]. Despite this specialization, individual DDR pathways also have an impressive ability to substitute for each other [10]. Together, the DDR minimizes the mutagenic impact of challenges during mitosis and after DNA damage. The importance of the DDR is best illustrated by the striking increases in tumorigenesis that result from deficiencies in one or more of its member pathways [11,12,13].

Understanding how the DDR functions and how defects in the DDR impact genomic integrity has direct implications for improving chemotherapeutics, countering chemoresistance, and genome editing. As a result, there is widespread research interest in the DDR from a diverse set of perspectives. This review discusses how a protein of beta genus of human papillomaviruses (β-HPV) promotes destabilization of the cellular genome by hindering DDR pathways. β-HPVs are small, double-stranded DNA viruses, some of which may contribute to non-melanoma skin cancer in immunocompromised individuals [14,15]. They also commonly infect the general population. Their role in promoting oncogenesis is controversial. While numerous in vitro and in vivo studies have demonstrated oncogenic properties of certain β-HPV proteins [16], the absence of β-HPV gene expression in tumors has caused many to question the physiological relevance of these properties regarding oncogenesis. Further, a recent report suggests that cutaneous HPVs protect against skin carcinogenesis [17]. This controversial claim resulted in an interesting series of articles where the evidence for and against β-HPV infections protecting against skin cancer development was discussed [18,19].

To complete their lifecycle, β-HPVs require skin cells to remain proliferatively active, despite stimuli (e.g., differentiation and UV exposure) that are known to induce cell cycle exit. The E6 protein from β-HPVs (β-HPV E6) plays a notable role in promoting this aberrant proliferation. All five β-HPV species (beta-1, beta-2, beta-3, beta-4, and beta-5) encode E6 [20]. β-HPV E6 is a small, 150 amino acid protein that contains two zinc-finger domains at its N- and C-termini [21]. The β-HPV E6 from some β-HPVs (e.g., HPV5 and HPV8) binds and destabilizes a cellular histone acetyltransferase, p300, to counter the cell cycle arrest and apoptosis associated with differentiation and DNA damage [22]. It is currently unclear how common p300 binding by β-HPV E6s is because only a few β-HPV types have been thoroughly studied; however, p300 destabilization is not a universal feature of β-HPV E6s. Among the β-HPV E6s that have been well characterized, p300 binding varies from weak/no attachment to robust tethering [23,24]. This review discusses these topics and others, with a particular focus on HPV8 E6. β-HPV E6 functions not related to p300 binding have been extensively reviewed [25,26,27,28] and will not be discussed in this review.

## 2. Materials and Methods

The E6 amino acid sequences from 53 beta genus of human papillomaviruses available at PaVE (https://pave.niaid.nih.gov, accessed on 14 May 2021) were downloaded and aligned. A phylogenetic tree was constructed using PaVE.

## 3. Results

### 3.1. p300 Is a Master Transcription Regulator and Tumor Suppressor

A great deal has been determined about p300 through traditional approaches geared toward dissecting its biology. It is encoded by the gene *EP300* and belongs to the type 3 family of lysine acetyltransferases, with homologs found in mammals and other multicellular organisms such as flies, worms, and plants [29,30]. p300 consists of conserved domains, including a central catalytic domain (KAT) responsible for protein acetylation that is adjacent to a bromodomain and PHD finger (CH2), both of which aid in chromatin association and modification [31,32]. The central domain is flanked by four transactivation domains: (i) the cysteine–histidine-rich region 1 (CH1) that contains a transcriptional adapter zing finger 1 (TAZ1), (ii) a kinase-inducible interacting domain (KIX), (iii) another cysteine–histidine-rich region (CH3) that includes a TAZ2 and a ZZ domain that are known to interact with a wide range of proteins including HPV8 E6, and (iv) a nuclear receptor coactivator binding domain (IBiD) (Figure 1) [33]. As a transcriptional coactivator, it interacts with over 400 factors, allowing it to regulate physiological processes including the DDR, differentiation, proliferation, and apoptosis [34,35]. p300 also acts as a stabilizing scaffold between transcription machinery and transcription factors that bind through the CH1, CH3, and KIX domains of p300 [36,37]. p300 promotes transcription directly by acetylating histones through its HAT domain [38,39]. It also acetylates non-histone proteins and, as a result, modifies their activities [40].

While these activities have a wide ranging ability to alter cellular processes, they are closely linked to promoting the DDR [41,42,43]. For example, p300 contributes to the homologous recombination pathway by transcriptionally activating *RAD51* and *BRCA1* [44]. Additionally, loss of p300 lead to defects in cell cycle arrest induced by DNA replication errors and is correlated with a lack of CHK1 phosphorylation [45]. Cell lines harboring *EP300* mutations display faulty base excision repair in response to oxidative damage [46]. Suppression of p300 histone acetyltransferase activity significantly abrogates the recruitment of DDR factors normally seen in response to UV [47]. Additionally, p300 promotes pRB, p53, and TGF-β signaling [48,49,50,51].

Given the established tumor suppressor role of the DDR and its other targets, it is not surprising that p300 mutations are associated with tumorigenesis and act as a tumor suppressor in cutaneous squamous cell carcinoma (cSCC) [52]. *EP300* mutations, largely missense point mutations, are found across a wide variety of cancer types [53]. However, mutations were most frequent in cSCC as reported in the COSMIC (Catalogue of Somatic Mutations in Cancer) database [54]. Further, decreased nuclear p300 staining is associated with disease progression in melanoma patients [55]. Finally, a heterozygous germline *EP300* mutation results in reduced p300 abundance that manifests as Rubinstein–Taybi syndrome (RSTS), a condition characterized by increased cancer predisposition [56].

### 3.2. HPV8 E6 Reduces Genome Stability by Destabilizing p300

Although multiple β-HPV E6 proteins bind and destabilize p300, HPV8 E6 is the most studied β-HPV E6. Further, because HPV8 infections are hypothesized to promote non-melanoma skin cancer by increasing the mutagenic risk of UV exposure, the ability of HPV8 E6 to hinder the repair of DNA damaged by UV has been a research focus. As a result, characterizations of HPV8 E6 biology have complemented and confirmed more traditional molecular biology studies, dissecting the role of p300 in preserving genome stability. The following sections highlight the known mechanisms by which HPV8 E6-mediated p300 destabilization hinders genome stability.

#### 3.2.1. ATR

ATR is the principle kinase activated in response to UV and replication stress [57]. By destabilizing p300, HPV8 E6 reduces ATR mRNA and protein levels [58]. This demonstrates a role for p300 in ATR transcription, presumably through histone modification. The reduction in ATR abundance results in delayed formation of ATR repair complexes following UV damage. As expected, HPV8 E6 delays the resolution of UV-photolesions and increases the frequency of UV-induced double-stranded breaks in DNA (DSBs). While these studies were conducted in cell line models, similar effects were seen in transgenic mice expressing HPV8 E6 from the K14 promoter. Namely, HPV8 E6 made UV-damage more persistent, made UV-induced DSBs more common, and diminished ATR activation [59]. These repair defects are likely the result of impaired ATR signaling, as HPV8 E6 attenuates the phosphorylation of ATR and its downstream targets CHK1 and CDC25A [59,60]. Unlike the p300-dependent reduction in ATR by HPV8 E6, the role of p300 destabilization in downstream ATR signaling has not been demonstrated. HPV8 E6 also impaired ATR-dependent events critical for the cellular response to UV damage (nuclear localization of XPA and pol η repair complex formation) but, again, the requirement of p300 destabilization in these hindrances was not tested [60,61,62,63].

#### 3.2.2. ATM

ATM is one of the primary kinases responsible for orchestrating the cellular response (i.e., apoptosis, cell cycle checkpoint activation, and DNA repair) to DSBs through phosphorylation of its hundreds of targets [57,64]. HPV8 E6 decreases total ATM protein levels by destabilizing p300 [65]. Following UV damage in cell culture models, HPV8 E6 also diminishes ATM activation (via autophosphorylation) and reduces ATM-mediated phosphorylation of two essential downstream DDR factors, BRCA1 and CHK2 [60]. Further, HPV8 E6 reduces UV-induced ATM phosphorylation in three-dimensional organotypic raft cultures [59]. As in the repression of ATR signaling by HPV8 E6, while p300 destabilization is the most likely mechanism by which HPV8 E6 attenuates downstream ATM signaling, this has not been demonstrated [60,66].

#### 3.2.3. DNA-PK

DNA-PK is the other principal kinase activated in response to DSBs. DNA-PK is unique compared to ATM and ATR, in that it is a holoenzyme, made up of DNA-dependent protein kinase catalytic subunit (DNA-PKcs) and a heterodimer, of Ku70 and Ku80 [67]. DNA-PK is recruited to DSBs by 53bp1, leading to its autophosphorylation, and ultimately to the phosphorylation of downstream proteins, including Artemis [68]. HPV8 E6 delays the resolution of DNA-PKcs and 53bp1 foci by destabilizing p300 [69,70]. HPV8 E6 also reduced phosphorylation of DNA-PKcs and Artemis [69].

#### 3.2.4. Double Strand Break Repair

The reduction in ATM and DNA-PKcs signaling suggests that HPV8 E6 impairs DSB repair. Homologous recombination (HR) is the primary mechanism for DSB repair, while non-homologous end joining (NHEJ) generally serves as a backup repair mechanism when HR fails [71,72]. HR is usually restricted to cell cycle segments when a homologous template is available (i.e., S and G2 phase), while NHEJ is not restricted to any particular portion of the cell cycle [73,74]. HPV8 E6 delays the repair of DSBs by reducing the efficacy of HR and NHEJ. In both cases, the mechanism of action by which HPV8 E6 attenuates repair is the destabilization of p300. For the HR pathway, destabilization reduces the abundance of p300 at the promoters of two essential HR genes, *BRCA1* and *BRCA2* [70]. As a result, there are fewer *BRCA1* and *BRCA2* transcripts, lower protein abundance, and fewer repair complexes formed in response to DSBs [70]. Despite BRCA1 and BRCA2 facilitating an earlier step in the HR pathway, HPV8 E6 does not prevent RAD51 foci formation [70]. However, the RAD51 foci that form are not resolved, suggesting that the reduction in p300 results in further deregulation of the HR pathway. HPV8 E6 also attenuates the repair of DSBs by NHEJ. NHEJ-requires DNA-PK activity. As described in the previous section, by destabilizing p300, HPV8 E6 hinders DNA-PK autophosphorylation and the phosphorylation of at least one DNA-PK substrate (Artemis). The reduced p300 abundance also delays the resolution of at least two NHEJ repair complexes (DNA-PKcs and 53bp1) [69,70].

#### 3.2.5. Cell-Cycle Checkpoints and Differentiation

Growth arrest is a powerful tool for responding to threats to genome stability [75,76,77]. Preventing growth allows time for repair to occur or missegregated chromosomes to resolve. If the danger is not mitigated, permanent growth arrest assures that cells containing a mutated genome do not propagate. Additionally, differentiation was recently identified as a mechanism by which polyploidy epithelial cells are prevented from growth [78]. Despite the benefits to the cell, pauses to cellular growth run counter to the requirements for β-HPV replication.

The E6 from multiple types of β-HPVs inhibits the cues that normally stop cells from dividing upon UV exposure [79,80]. However, the role of p300 destabilization in escape from UV-induced cell cycle arrest was not tested in these studies. p300 destabilization allows HPV8 E6 to promote proliferation in the face of other genome destabilizing events and reduces differentiation both in vivo and in vitro [81,82,83,84,85]. One mechanism by which reduced p300 availability promotes proliferation is the dysregulation of the Hippo pathway. By destabilizing p300, HPV8 E6 increases the expression and protein levels of pro-proliferative Hippo pathway factors: CTGF, AXL, and SERPINE1 [82]. Altered Hippo pathway signaling has genome destabilizing consequences, as it allows cells to continue proliferating after failed cytokinesis, leading to viable cells with polyploid genomes [84]. Similarly, HPV8 E6 allows cells with more than three centrosomes to continue to proliferate by destabilizing p300, but a connection to the Hippo pathway has not been examined [84].

HPV8 E6 also reduces differentiation by destabilizing p300 [83]. This occurs at least in part through the downregulation of CCAAT/enhancer-binding protein α (C/EBPα), another p300-responsive gene (24). C/EBPα is a pro-differentiation transcription factor and, by decreasing its availability, HPV8 E6 causes a reduction in the differentiation marker involucrin. The reduced C/EBPα suppresses its transcriptional activator activity on microRNA-203 (miR-203), resulting in increased proliferation. Other differentiation markers (i.e., K1 and K10) are also reduced by HPV8 E6-mediated p300 destabilization, possibly as a result of C/EBPα reduction [83].

#### 3.2.6. p53

p53 is the most commonly mutated tumor suppressor found in human cancer [86]. By acting as a transcription factor, it regulates DNA repair, cell cycle arrest, senescence, angiogenesis, apoptosis, and many other pathways [87,88,89,90]. Unlike high-risk alpha-HPVs, all but one β-HPV E6 (HPV49 E6) are unable to bind and degrade p53 [91,92]. However, HPV8 E6 attenuates p53 signaling by destabilizing p300. p53 binding to chromatin regulates p53 activity in response to DNA damage [93]. p300 facilitates this damage-induced response by acetylating both histones and p53 [48,94,95]. HPV8 E6 reduces p53 acetylation (K382) in response to UVB exposure [58]. p53 is also regulated by ATR via phosphorylation on its serine-15 and -37 residues [96,97]. These post-translational modifications inhibit MDM2-mediated degradation stabilizing p53. This allows p53 to accumulate, prompting activation of DDR genes and association with p300 [98,99]. Likely by destabilizing p300, HPV8 E6 lessens phosphorylation of p53 at Serines 15 and 37 in response to UV, and limits p53 accumulation [58]. However, the role of p300 destabilization in p53 phosphorylation has not been tested.

The inhibition of p53 signaling by HPV8 E6 is not restricted to the response to damaged DNA. HPV8 E6 also reduces p53 accumulation in response to mitotic errors [84]. The destabilization of p300 by HPV8 E6 lowered p53 levels in the binucleated cells that form as a result of failed cytokinesis [84]. Stabilization of p53 in response to failed cytokinesis requires activation of the Hippo pathway, specifically LATS2 phosphorylation [100]. This causes LATS2 to bind p53 and inhibit MDM-mediated degradation [101,102]. HPV8 E6 reduces LATS2 activation and prevents p53 buildup induced by cytokinesis failure [82]. However, HPV8 E6 does not affect the activation of the Hippo pathway in response to high cell density. Although less mechanistically clear, the destabilization of p300 by HPV8 E6 also prevents p53 accumulation in response to the accumulation of supernumerary centrosomes [84]. Together, these data show that p300 protects against chromosomal instability [103,104,105].

### 3.3. How Conserved Is p300 Binding among β-HPV E6s

Only a handful of the 53 β-HPVs have been studied for their biological function in vitro or in vivo. As a result, little is known about the p300 binding potential of most β-HPV E6 proteins. To this end, the review has focused exclusively on HPV8 E6. However, four other β-HPV E6 proteins (HPV 5, -20, -25, and -38 E6) have been shown to bind p300 by via immunoprecipitation, followed by immunoblot and/or mass spectrometry [83,106,107]. Of these, HPV5 E6 and HPV38 E6 have been the most thoroughly characterized. HPV5 E6 behaves similarly to HPV8 E6 in as much that it binds and destabilizes p300 [83]. As both HPV5 E6 and HPV8 E6 destabilize p300, it is reasonable to assume that the p300 degradation-dependent phenotypes discovered in one apply to the other. Supporting this idea, HPV5 E6 and HPV8 E6 share many properties. These include: (i) destabilizing p300 [83], (ii) increasing thymine dimer persistence, (iii) increasing DSB prevalence after UVB exposure [58], (iv) decreasing ATR expression and activity [58], (v) reducing post-translational modifications of p53 [58], (vi) reducing ATM protein levels [65], (vii) increasing the frequency of cells with more than two nuclei, (viii) increasing the frequency of cells with more than 4N DNA, (ix) decreasing senescence-associated β-galactosidase levels in late passage primary cells [84], (x) increasing the average number of centrosomes per cell [84], (xi) attenuating p53 signaling in response to failed cytokinesis and supernumerary centrosome [84], and (xii) reducing *BRCA1*/*BRCA2* expression [70]. It should be noted that HPV8 E6′s p300 degradation, but not binding, is cell line dependent, as cell lines can harbor a constitutively active AKT [83]. Other groups have also failed to show p300 destabilization in some cell lines, suggesting that there may be additional factors that determine whether HPV8 E6 destabilizes p300 [59].

HPV38 E6 binds p300 less stringently than HPV5 E6 or HPV8 E6. As a result, HPV38 E6 does not appreciably destabilize p300. However, it impacts p300 signaling. For example, HPV38 E6 reduces p53 acetylation at lysine 382 [85]. In addition, p300 binding by HPV38 E6 is likely necessary for HPV38 E6 and E7 to immortalize primary cells [85]. The connection between HPV38 E6 binding p300 and immortalization relies on a mutation in HPV38 E6, and such mutants can produce data that are difficult to interpret (limitations and concerns with β-HPV E6 mutants are discussed further below). While HPV38 E6 shares some phenotypes with HPV8 E6, immortalization is reasonably unique to HPV38 E6, as neither HPV5 E6 nor HPV8 E6 immortalize cells with or without their associated E7 protein. This could be due to HPV38 E6 binding p300 with a different portion of its E6 protein than HPV5 E6 and HPV8 E6. Thus, HPV38 E6 may impact a different subset of p300-dependent signaling events.

Other β-HPV E6 proteins are predicted to destabilize p300 based on amino acid alignment to β-HPV E6s known to destabilize p300 (Figure 2). The p300 binding domains of HPV5 E6 or HPV 8 E6 are conserved in eight other β-HPV E6s (HPV12, -14, -19, -47, -99, -10, -143, and -203 E6). All of these except HPV206 are members of one of the most prevalent genera of β-HPVs (beta-1 genus), along with HPV5 and HPV8 [108]. HPV206 is currently unclassified [20,108]. Notably, HPV12, -14, and -47, along with HPV5 and HPV8, are associated with skin lesions in people with the rare genetic disorder epidermodysplasia verruciformis [109,110,111]. HPV47 E6 and HPV14 E6 also share significant amino acid sequences with HPV5 E6 and HPV8 E6, beyond the residues required for p300 binding [112]. The *Mus musculus* papillomavirus 1 (MmuPV1) is used as an in vivo model for β-HPV, as it shares some biological and biochemical properties with HPV8 E6 [113]. However, MmuPV1 E6 does not bind p300 [114] and does not contain a conserved p300-binding sequence (Figure 2). Therefore, MmuPV1 is unlikely to fully reproduce the biology of the β-HPVs most closely linked with epidermodysplasia verruciformis. An in vivo model system using cottontail rabbit papillomavirus (CRPV) DNA injected into rabbits demonstrated that CRPV E6 mutants deficient in binding p300 did not induced tumor formation, nor did they prevent apoptosis [85]. However, unlike the MmuPV1 model, only DNA of the virus was used instead of an infectious-replicative virus.

### 3.4. Confirming the Dependence of a HPV8 E6 Phenotype on p300 Destabilization

While HPV8 E6 clearly alters cell signaling by destabilizing p300, some caution needs to be exercised when determining whether a phenotype is the result of p300 destabilization. The initial screening for identifying phenotypes caused by HPV8 E6-mediated reductions in p300 is typically performed by comparing a vector control cell lines to ones expressing either wild type HPV8 E6 or HPV8 E6, with the residues responsible for p300 binding deleted or mutated (e.g., HPV8 E6Δ132-136). Phenotypes seen in cells expressing wild type HPV8 E6 but reduced or absent in the HPV8 E6Δ132-136 are potentially the result of p300 destabilization. However, data from the HPV E6 mutants can easily be misinterpreted because deletions and mutations in the small viral protein frequently disrupt more than one aspect of HPV E6 biology. This is also the case for HPV8 E6 mutants. For example, deletion of the residues of HPV8 E6 that facilitate p300 binding (Δ132-136) also prevent HPV8 E6 from binding to MAML1, which is critical for HPV8 E6 inhibition of the NOTCH pathway [114]. As a result, comparisons between wildtype and mutant HPV8 E6 cannot distinguish between a phenotype that is the result of p300 destabilization, MAML1 binding, or attributable to a more generalized loss of HPV8 E6 function. This is particularly true for phenotypes that are only reduced (rather than abrogated) in cells expressing HPV8 E6Δ132-136. Table 1 lists phenotypes found in cells expressing HPV8 E6 that remain partially intact in cells expressing HPV8 E6Δ132-136. While this gives reason for concern that the HPV8 E6Δ132-136 mutant might cause entirely non-specific inhibition of HPV8 E6, a subset of phenotypes is not reduced when these residues are deleted. Table 2 contains a list of these phenotypes. Therefore, HPV8 E6Δ132-136 is not an entirely functionless protein.

More targeted mutations to the p300-binding site are needed to eliminate the concern associated with HPV8 E6Δ132-136. However, no consensus has been reached on the efficacy of the currently available alternatives. For example, there are conflicting reports over the extent to which HPV8 E6K136N is deficient for p300-binding [59,114]. These differences may be attributable to differences in experimental approaches. HPV8 E6K136N failed to bind p300 when transiently expressed in RTS3b cells, but continued to bind p300 when stably expressed in U2OS cells [59,114]. One other mutation (HPV8 E6H135A) was initially described as having a significantly reduced ability to bind p300 [117]. Unfortunately, the HPV8 E6H135A mutant has not been thoroughly characterized (or used) since it was initially described.

Currently, the best way to validate the p300 dependence of any phenotype caused by β-HPV E6 is to use other molecular techniques as complimentary approaches. To this end, cells expressing β-HPV E6 have been transiently transfected with either phosphomimetic (p300 S1834E) or phospho-dead (p300 S1834A) p300 mutants [58,65,70,83]. Since HPV8 E6 reduces p300 abundance by blocking its phosphorylation by AKT, the phosphomimetic p300 mutant is resistant to HPV8 E6-mediated destabilization [83]. p300 S1834A is catalytically inactive and thus serves as a negative control [118]. In this system, phenotypes that require p300 destabilization are lost in cells transfected with p300 S1834E and maintained in cells transfected with p300 S1834A. A disadvantage of using this approach to confirm the p300 dependence of a phenotype is the transient nature of p300 mutant expression that limits the types of phenotype that can examined.

The role of p300 destabilization in phenotypes can also be confirmed using non-viral mechanisms of knocking p300 down or out. siRNA-mediated knockdown of p300 has been used to validate cellular process identified as likely requiring p300 destabilization [58,65,81,83,85]. Since the knockdown of p300 is transient, this approach is also not ideal for testing phenotypes associated with prolonged HPV8 E6 expression. shRNA may be useful for testing the p300-dependence of phenotypes that take more time to occur. However, RNAi-mediated degradation can result in off-target effects or require multiple siRNAs/shRNAs to be pooled together to reach a significant reduction in p300, further enhancing possible off-targets.

Genetic knockouts have the potential to eliminate these concerns and have been used to test the role of p300 in multiple HPV8 E6-mediated phenotypes [69,82]. Most commonly, colorectal cancer cell lines (HCT116) with and without the p300 gene knocked out are compared [119]. A clear disadvantage of this approach is that the cell type is unlikely to be physiologically relevant to β-HPVs. In the future, it would be useful to knockout the p300 gene in primary or immortalized keratinocytes to overcome this issue.

Small molecule inhibitors of p300 have been developed and offer the ability to impair p300 activity in keratinocytes [120,121,122]. Our unpublished data have confirmed that p300 inhibitor CCS1477 is capable of reproducing aspects of HPV8 E6 biology that require p300 destabilization. One potential advantage of using p300 inhibitors is their ability to target the protein directly. However, because p300 would not be destabilized by a small molecule inhibitor, it could still act as a scaffold. Finally, an untested yet feasible option is to use Rubinstein–Taybi syndrome (RSTS) patient-derived cell lines that have reduced p300 levels and increased genomic instability, attributable to defective DNA repair [46,123].

The concerns about HPV8 E6 mutants necessitate confirming the p300 dependence of any phenotype with these or other approaches. As noted in this section, each of these methods has its limitations. As a result, using multiple confirmatory approaches raises confidence that HPV8 E6 is causing a given phenotype by destabilizing p300. Table 3 lists the phenotypes described in this review and the methods that were used to confirm their dependence on p300 destabilization by HPV8 E6. To the best of our knowledge, the role of p300 in these phenotypes was first identified by studying HPV8 E6.

## 4. Discussion

In this review, we aligned β-HPV E6 proteins based on their amino acids. This grouping shows that E6 proteins with proven or putative p300 binding capability cluster together. The extent to which these β-HPV E6 proteins represent a group with unique biology has yet to be fully determined. Similarly, the extent to which viral life cycle differences exist between β-HPVs that express E6s that can and cannot destabilize p300 has not been explored. 

## 5. Conclusions and Future Directions

As a transcription factor, p300 regulates the expression or activity of hundreds of genes directly. Many of those genes (e.g., p53, ATM, and ATR) are key factors in other signaling pathways where they control the expression or activity of multiple other genes. This broad influence makes p300 a critically important factor in maintaining genomic stability via the DDR, cell cycle checkpoints, the regulation of p53, and likely other mechanisms. Unsurprisingly, aberrant p300 biology is linked to multiple cancers and disease states [46,53,124,125]. Therefore, it is important to examine p300 biology from multiple angles. Investigations into HPV8 E6 biology represent a markedly different approach. While studies of HPV8 E6 biology are generally not designed to learn about p300, they have nevertheless provided significant insights into p300 biology (Table 3). As well as the potential to learn about p300 by studying HPV8 E6, it is important to note that multiple control and confirmatory experiments are required before strong conclusions about p300 can be made. Furthermore, there are accumulating data that HPV8 E6 reduces the cell’s ability to address errors during mitosis, such as failed cytokinesis or centrosome duplication [82,84]. As a result, the genome-destabilizing potential of HPV8 E6 extends beyond the inhibition of UV-damaged DNA.

There are several opportunities to expand our understanding of how the binding of p300 by β-HPVs impacts viral biology. First, sequence alignments suggest that the number of β-HPVs that deregulate p300 activity is higher than what has been demonstrated. These β-HPVs are therefore also likely to impair genomic stability, but this should be confirmed experimentally. Next, it is critical to determine if the disruption of p300 signaling is heightened, diminished, or unaltered by other β-HPV early proteins (i.e., E1, E2, E7, E8^E2) that are expressed alongside β-HPV E6 during naturally occurring infections. Among these, β-HPV E7 demonstrates potential synergy with E6 to disrupt the DDR via its ability to attenuate expression of DDR- and apoptosis-related genes [126] Further, some β-HPV E7s bind and destabilize the tumor suppressor pRB, which could promote UV-damaged cells to bypass cell cycle checkpoints [91,127].

Finally, HPV8 E6 may alter CREB-binding protein (CBP) signaling. CBP shares significant sequence homology with p300 and is also a critical regulator of RNA polymerase II-mediated transcription via histone acetylation [34,128]. Despite their high level of homology, p300 and CBP regulate distinct gene sets and thus are not entirely redundant [129]. It was hypothesized that all mammalian E6 proteins may interfere with CBP/p300 functions through direct interaction or by capturing LxxLL containing CBP/p300 partners [130]. Driving this idea is mammalian E6′s ability to recognize LxxLL motifs and CBP/p300’s critical role in numerous functions, including a host’s innate antiviral response [130,131]. To our knowledge, the only available data regarding HPV8 E6 and CBP come from two separate pulldown mass spectrometry experiments that found CBP peptides to be associated with HPV8 E6 [83,106]. Despite the potential interaction, HPV8 E6 did not reduce CBP levels [83]. While this suggests that HPV8 E6 may not degrade CBP, studies of other viral proteins provide reason to believe the interaction could be significant. The adenoviral protein E1A alters CBP signaling in order to promote S-phase by binding (but not destabilizing) CBP [125,132,133,134]. Given that CBP is a master transcription regulator, akin to p300, dysregulation of CBP signaling would be expected to markedly alter cellular environment.

Clearly, HPV8 E6 did not evolve the ability to destabilize p300 to serve as a tool for gaining insight into DNA repair and cell signaling. Instead, the destabilization of p300 likely provides the virus a replicative advantage. The damaging effect of UV exposure on the skin normally induces cell cycle exit. This would oppose HPV8 replication, as β-HPVs require actively proliferating cells to complete their lifecycle. The destabilization of p300 affords HPV8 the opportunity to attenuate the cell cycle arrest associated with UV and other genome destabilizing events. It also increases the expression of pro-proliferative genes and inhibits differentiation.

## Figures and Tables

**Figure 1 viruses-13-01662-f001:**
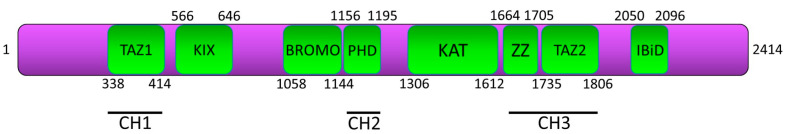
p300 structural domains and their localization. TAZ1 (also known as CH1), KIX, BROMO, PHD (also known as CH2), KAT, ZZ and TAZ2 (together known as CH3), and IBiD. Approximate domain boundaries were taken from the p300 Pfam database entry (Q09472).

**Figure 2 viruses-13-01662-f002:**
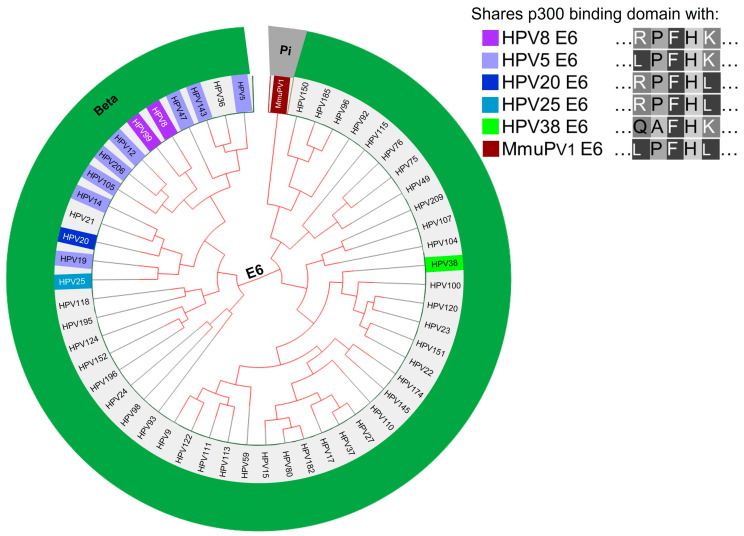
Phylogenetic tree of β-HPV and MmuPV1 based on the E6 nucleotide sequence (https://pave.niaid.nih.gov/, accessed on 14 May 2021). E6s with identical amino acid sequence highlighted to match corresponding E6s either known to bind p300 (HPV8, 5, 20, 25, 38) or not bind p300 (MmuPv1).

**Table 1 viruses-13-01662-t001:** Phenotypes of HPV8 E6Δ132-136 that are weaker than those of wild type HPV8 E6. Cell lines abbreviations: primary human epidermal keratinocytes (PHEK), human foreskin keratinocyte (HFK). Stable expression via lentiviral transduction.

Cell Type	Expression	Partial Phenotype	Reference
PHEK	Stable	Reduction in Syntenin-2 mRNA and protein levels	[115]
HFK	Stable	Increased persistence of thymine dimers after UVB	[58]
HFK	Stable	Augmented late passage cells with >4 N content	[84]
HFK	Stable	Fewer BRCA1/2 positive cells after IR	[70]
HFK	Stable	Increased sensitivity to PARP1 inhibitor	[70]
HFK	Stable	Delays RAD51 foci resolution after 4 gray of IR	[70]
HFK	Stable	Enhances sensitivity to IR	[70]

**Table 2 viruses-13-01662-t002:** Phenotypes of HPV8 E6Δ132-136 analogous to wild type HPV8 E6. Cell lines abbreviations: human osteosarcoma cell line (U2OS), human foreskin keratinocytes (HFK), HPV-negative cervical carcinoma cell line (C33A), HPV-negative skin squamous cell carcinoma-derived cell line (RTS3b), primary human foreskin keratinocytes (NHK). “Stable” denotes stable expression achieved via lentiviral transduction. “Transient” denotes transient expression achieved via transfection.

Cell Type	Expression	Phenotype	Reference
U2OS	Stable	Inhibits non-homologous end joining	[69]
C33A	Transient	Precipitated with PTPH1	[116]
RTS3b	Transient	Activates early HPV8 promotor	[117]
NHK	Stable	JunB mRNA expression downregulated	[81]
HFK	Stable	Diminishes senescence-associated β-galactosidase staining in late passage binucleated cells	[84]
HFK	Stable	Increases frequency of cell with three centrosomes	[84]
HFK	Stable	Increases growth rate in late passage cells	[84]
U2OS	Stable	Prevented XRCC4 foci in response to DSBs	[69]
U2OS	Stable	Decreases H_2_O_2_-induced DNA-PKcs phosphorylation	[69]

**Table 3 viruses-13-01662-t003:** Phenotypes of HPV8 E6 that have been confirmed to be through the destabilization of p300. p300 mutants include S1834E and S1834A. E6 mutant refers to HPV8 E6Δ132-136. Knockout and Knockdown are via siRNA and HCT cells without p300, respectively.

Phenotype	p300 Activity Confirmed via:				Reference
	p300 Mutants	E6 Mutant	Knockout	Knockdown	
Reduction in K1, K10, and involucrin mRNA expression		×		×	[83]
Diminish ATR expression	×	×		×	[58]
Lessen ATM expression	×	×		×	[65]
Inhibit p53 accumulation in binucleated cells	×	×			[84]
Prevent p53 buildup in cells with ≥3 centrosomes	×	×			[84]
Allow binucleated cells to proliferate	×	×			[84]
Allow cells with ≥3 centrosomes to proliferate	×	×			[84]
Reduce BRCA1 and BRCA2 expression		×	×		[70]
Attenuate DSB repair	×	×	×		[70]
Reduce C/EPBα and miR-203 expression	×			×	[81]
Upregulate expression of pro-proliferative Hippo pathway genes		×	×		[82]
Attenuate DNA-PKcs phosphorylation after DSB induction		×	×		[69]
Decrease phosphorylation of Artemis after DSB induction		×	×		[69]

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
