# Peer review of "Beta-Genus Human Papillomavirus 8 E6 Destabilizes the Host Genome by Promoting p300 Degradation"

_viruses, 2021, doi:10.3390/v13081662_

Round 1

Reviewer 1 Report

This is an interesting paper focusing on the E6 protein of the beta HPV8 and the consequent effects on the host genome stability through destabilization of p300.  The paper is well written and summarizes well the specific interests of the key author.  The functions of the oncoprotein encoded by the beta HPVs are a bit confusing given the diversity of cellular proteins engaged by the different HPV types (aside from the MAML1). This review provides a good platform for studying these various E6 proteins and their different activities.  Although the focus is on groomer destabilization, DNA repair pathways and cancer, an etiologic role of any of the beta HPVs remains a question and should be highlighted in this review.  Further some discussion of the recent work of Strickly et al (Nature 575:519, 2019) might help balance the review discussion.  Finally the authors should be reminded that the principle activities of the viral proteins lie in the replication of the viruses themselves and creating a cellular environment to promote such replication.  Some discussion of how the destabilization of p300 may contribute to such would benefit the review.

Reviewer 2 Report

In this review the authors aimed at summarizing our current knowledge on the interaction of betapapillomavirus E6 proteins with the cellular protein p300. The review addresses a subject that is of great interest to the scientific community. The work is well-structured, well written and deserves publication in this journal. However, the following minor points need to be considered:

  • Line 43: the authors should not put “differentiation” as external stimuli. Please rephrase.
  • Lines 56 and 60: The use of subheadings in review is unusual. Please check.
  • Section 3.2: the authors should state somewhere, may be in this section, that p300 acts as a tumor suppressor in the skin.
  • Line 119: should be “cell line models” and not “cell models”
  • Lines 204-205: the authors should state that beta-HPV E6s do not bind p53 with references and then state that HPV49 E6 is able to do it.
  • Section 3.3: the authors may need to state that other groups also see binding of E6 to p300 but did not detect p300 destabilization.
  • Do the authors really need to include the section on MmuPV1 in respect to its binding ability to p300? They should rather consider Muench et al., 2010, Cancer Res. in which a central role of p300 in skin tumorigenesis is shown in a papillomavirus animal model.
  • Discussion: at the very end of the discussion section it would be nice to have a final paraph on the role of the E6 - p300 interaction for viral life cycle. They should also consider the speculations published by Dr. Trave here (Suarez and Trave, 2018, Viruses, section 4.2 in the paper).

Reviewer 3 Report

The review by Dacus and Wallace provides a well-written and balanced discussion of the activities of HPV 8 E6 in inducing genetic instability through its effects on p300. The review provides a useful overview of the b-HPV proteins and describes their multiple activities as well as differences. The references are appropriate and the tables helpful

Some minor points:

1). It would be useful to expand the comparison of b-HPV E6 proteins with HR-E6 and their effects on DNA damage repair pathways. In HR-PVs E7 seems to be the major mediator of this response. Is anything known about how b-HPV E7 proteins act on these pathways?

2). An expanded discussion of the interplay of E6 targets p300 and MAMAl1 would be useful. Which activities of MMAl1 are critical and how does E6 affect these?
